# *Lepisanthes alata* Attenuates Carrageenan-Induced Inflammation and Pain in Rats: A Phytochemical-Based Approach

**DOI:** 10.3390/ph18081142

**Published:** 2025-07-31

**Authors:** Elvy Suhana Mohd Ramli, Nadia Mohamed Tarmizi, Nur Aqilah Kamaruddin, Mohd Amir Kamaruzzaman

**Affiliations:** 1Department of Anatomy, Faculty of Medicine, University Kebangsaan Malaysia, Jalan Yaacob Latif, Bandar Tun Razak, Cheras, Kuala Lumpur 56000, Malaysia; elvysuhana@ukm.edu.my (E.S.M.R.); aqikama05@gmail.com (N.A.K.); 2Anatomy Unit, Preclinical Department, Faculty of Medicine & Defence Health, Universiti Pertahanan Nasional Malaysia, Sungai Besi, Kuala Lumpur 57000, Malaysia; nadiatarmizi@upnm.edu.my

**Keywords:** *Lepisanthes*, inflammation, anti-inflammatory, carrageenan, acute pain, analgesic

## Abstract

**Background**: Inflammation abrogates cellular organization and tissue homoeostasis, resulting in redness, swelling, heat, pain, and loss of function. A model of carrageenan-induced paw edema (CIE) is commonly utilized to test anti-inflammatory substances. Based on the ability of *Lepisanthes alata* (LA), a tropical plant that is rich in phytochemicals like polyphenols, this study assessed the optimal dose and the health benefits of LA in rats that had been induced with carrageenan to develop paw swelling. **Methods**: Twenty-four male Wistar rats were divided into four groups to which carrageenan was administered, after which, distilled water at oral dose (C + DW), sodium diclofenac 25 mg/kg (C + DS), LA extract in 250 mg/kg (C + LA250), and 500 mg/kg (C + LA500) was given, respectively. Paw edema was assessed in 24 h. Pain was assessed using the Rat Grimace Scale (RGS), cytokines, antioxidant activity, and tissue changes. **Results**: LA at 250 and 500 mg/kg significantly decreased paw edema and inflammatory markers in the results of both studies. Remarkably, LA 250 mg/kg significantly decreased RGS scores as well as IL-1β, TNF-α, and histological inflammation but had a positive effect on T-SOD levels. Conclusions: LA extract, especially at 250 mg/kg, shows potent anti-inflammatory, analgesic, and antioxidant properties in CIE rats.

## 1. Introduction

Immune cells, blood vessels, and mediators respond to damage or potentially pathogenic stimuli with acute, local, complicated cellular and molecular inflammation [1]. Cells release cytokines, small signalling proteins that regulate inflammation and immunological responses [2]. Cytokines aid immune cell communication, direct them to infection or injury sites, and promote tissue healing during inflammation [3]. Influential pro-inflammatory cytokines in inflammation include TNF-α, IL-1, IL-6, and IL-17 [4,5]. TNF-α is a primary cause of acute inflammation. It makes blood arteries more permeable, helps immune cells reach injuries and infections, and produces pro-inflammatory cytokines [6]. By upregulating endothelial cell adhesion molecule expression, IL-1 promotes leukocyte migration into inflamed tissues and produces fever in the hypothalamus [7]. IL-6 regulates liver production of acute-phase proteins such C-reactive protein and immunological responses in acute and chronic inflammation [8]. Th17 cells release IL-17, which recruits neutrophils to fight bacterial and fungal infections [9]. Rheumatoid arthritis, inflammatory bowel disease, and psoriasis are characterized by chronic inflammation caused by excessive pro-inflammatory cytokines [10]. On the other hand, regenerative cytokines play a critical role in minimizing inflammation and promoting tissue healing, thereby preventing tissue destruction [11,12,13,14].

Inflammation and pain are closely linked, as inflammation induces pain via various biological mechanisms. Acute inflammation serves as a protective and restorative mechanism activated by injury or infection [15]. In this phase, pro-inflammatory mediators, including bradykinin, prostaglandins, and leukotrienes, are released, resulting in the activation and sensitization of nociceptors, which are specialized pain receptors. Mediators reduce the activation threshold of nociceptors, leading to hyperalgesia and heightened pain perception at the injury site [16,17]. Acute inflammation-related pain is typically transient and diminishes as the inflammation resolves. The resolution phase is facilitated by specialized pro-resolving lipid mediators, including lipoxins and resolvins, which antagonize pro-inflammatory signals and re-establish tissue homeostasis [18]. Interference with the natural resolution process, such as through the administration of anti-inflammatory drugs like NSAIDs or corticosteroids, may disrupt healing and potentially result in chronic pain. A study conducted by McGill University indicated that although these drugs relieve acute pain, they may impede the resolution of inflammation, thereby raising the risk of developing chronic pain states [19,20].

Despite their widespread use in managing acute pain, analgesics can significantly impact patient care due to their immediate side effects. The most common classes of analgesics include opioids, non-steroidal anti-inflammatory drugs (NSAIDs), and acetaminophen, each associated with specific adverse effects. While analgesics are necessary for acute pain management, they must be used with caution. Opioids provide considerable dangers, including respiratory depression and drowsiness, whereas NSAIDs can induce gastrointestinal and renal problems. Although acetaminophen is normally safer, it can cause liver damage. A multimodal approach can improve pain management while reducing the side effects associated with individual analgesics.

Herbal products have been traditionally used to manage inflammatory diseases due to their bioactive compounds, such as polyphenols, flavonoids, alkaloids, and terpenes, which modulate inflammatory pathways and alleviate symptoms [21]. Concerns over synthetic COX-2 inhibitors, such as rofecoxib and valdecoxib, emerged after these drugs were withdrawn by the U.S. FDA in 2004 and 2005, respectively, due to risks of cardiovascular and dermatological toxicities [22,23]. Herbal remedies are increasingly used as complementary treatments for chronic inflammatory diseases, including arthritis, inflammatory bowel disease, asthma, and skin conditions [23]. Notably, inhibitors of the 5-LOX enzyme derived from plants offer significant relief from chronic inflammation without reported adverse effects, making them a promising alternative therapy [15].

According to ethnobotanical studies, native people commonly consume certain Lepisanthes species as food sources and it is also applied in traditional medicine depending on the type of species and parts of the plant [24]. There are six species of the Lepisanthes genus that are used in traditional and folk medicine. They are *L. alata*, *L. amoena*, *L. fruticosa*, *L. senegalensis*, *L. rubiginosa*, and *L. tetraphylla*. They have been taken for years for one condition or the other, ranging from illnesses to symptoms like pain, dizziness, high fever, loose bowels, abscesses, healing of wounds, among others. The studies mentioned that the genus has many positive outcomes, including antioxidant [25,26], antihyperglycemic [27], antimalarial [28], analgesic [27], and anti-diarrheal effects [29].

*Lepisanthes alata* (LA) (Blume), also known as the Terengganu Cherry tree or Chinese Starfruit, belongs to the Sapindaceae family. In tropical and subtropical areas of Southeast Asia, this tree is widely distributed, and also grows naturally in Javanese, Sumatran, and Malay forests [30]. Malaysia cultivates the tree as an ornamental, yielding reddish, small, fleshy fruits akin to berries [29]. The younger of the two varieties of trees occurring commonly in Southern Thailand is said to supply edible young leaves that can be boiled and prepared as an accompaniment. It is used as an ornamental plant, as well as having ethnomedical applications and being a component of agroforestry systems [31,32]. While other species in the Lepisanthes genus, such as *L. senegalensis*, have shown cytotoxic properties due to specific triterpenes, LA does not exhibit such toxicity, highlighting its safety for consumption [25,33].

The model of paw edema induced by carrageenan serves as a prominent experimental framework for the investigation of acute inflammation and the assessment of the anti-inflammatory properties of diverse compounds. This model holds considerable importance owing to its reproducibility and sensitivity, enabling researchers to explore the fundamental mechanisms of inflammation and assess the effectiveness of prospective therapeutic agents. Meanwhile, the knowledge regarding the potential of LA in the inflammatory pain animal model has yet to be uncovered. Thus, in the present study, we investigated the anti-inflammatory activities of extracts obtained from the leaves of LA on Carrageenan-induced paw edema (CIE) using an in vivo experimental model.

## 2. Results

### 2.1. Radical Scavenging Activity and Total Phenolic Concentration

The aqueous extract of *L. alata* leaves contained a total phenolic content of 189.50 (mg GAE/g extract) where gallic acid was taken as a reference. The DPPH radical scavenging activity of the aqueous extract of *L. alata* yielded 55.65% inhibition.

### 2.2. Paw Edema LA Ameliorated the Inflammatory Effects of Carrageenan 1% Infiltration on Paw Edema

It was shown that the model group (C + DW) showed higher paw edema in all the recorded time intervals compared to the other group. Following carrageenan injection, the paw edema predominated on the model group starting at 2 h onwards (Figure 1).

Meanwhile, C + DS recorded significantly lower foot edema compared to the model group (C + DW) and averagely lower edema compared to both treatment groups (C + LA250) and (C + LA500) from the second until the first 6 h. This was evidenced by the significant difference in average foot edema (*p* < 0.05) between the C + DS group compared to the C + DW group at the second, fourth, and sixth hours with an average + SEM (6.55 + 0.16), (7.04 + 0.19), and (7.51 + 0.19), respectively. The percentage inhibition for the C + DS group was highest at 50.58% in the second hour. However, C + DS was unable to further reduce the paw edema 8 h onwards.

Despite the slower effect in the first 4 h, the C + LA250 group recorded a lower plantar edema compared to (C + DW) at 4, 6, 8, and 24 h with significant difference (*p* < 0.05) and recorded average + SEM of (7.83 + 0.30), (7.56 + 0.32), (7.03 + 0.37), and (5.82 + 0.27), respectively. At the 24th hour, it was found that the C + LA250 group recorded the lowest foot edema of rats compared to the C + DS group with a significant difference (*p* = 0.023) and an inhibition percentage of 63.25%.

The C + LA500 group also recorded lower foot edema compared to the C + DW group, at 4, 6, 8, and 24 h with significant differences of (*p* < 0.05) + SEM (7.50 + 0.14), (7.47 + 0.27), (7.12 + 0.27), and (6.09 + 0.18), respectively. In fact, beginning at the eighth hour onwards, both the C + LA250 and C + LA500 groups showed a longer lasting reduction in plantar edema compared to the C + DS group.

### 2.3. Effects of Rat Grimace Scale with Co-Treatment of LA Leaf Extract on Carrageenan-Induced Paw Edema in Rats

The score for the Rat Grimace Scale recorded an increase proportional to the time after the carrageenan injection. The RGS score of the model group (C + DW) was negative, reaching the average value of + S.E.M (1.0975 + 0.10845), which was the maximum as early as after 4 h. There was a significant difference between the C + DW group compared to C + DS group at the second until eighth hour with a value of *p* < 0.001 (Figure 2).

At the eighth hour, the maximum score was achieved and recorded for the three groups, namely the C + DS, C + LA250, and C + LA500 groups. The highest score at the eighth hour was recorded by the C + LA500 group (1.1942 + 0.07857) followed by the C + LA250 group (1.0283 + 0.04531). Meanwhile, the lowest score among the groups at the eighth hour was recorded by the C + DS group with an average score of + S.E.M (0.7342 + 0.04880).

At the end of the study, among all the participating groups, the lowest RGS score was recorded by the C + DS group while the highest score was recorded by the C + LA500 treatment group with an average score + SEM, 0.75+ 0.09.

### 2.4. Effects of Co-Treatment of LA Leaf Extract on White Cell Count of Carrageenan-InducedPaw Edema in Rats

The findings of the current study showed that LA leaf extract did not show significant effects for the concurrent treatment of carrageen-induced paw edema in rats (Figure 3). Instead of the expected elevation of differential count in the model group, the treatment group showed an increase in trends of white cells and neutrophil counts. It was further clarified that one-way ANOVA did not show any statistically substantial variation in all the findings of white cell count.

The highest white cell count was seen in the C + LA500 group (15.48 + 2.37 × 10^9^/L), followed by C + LA250 (14.35 + 2.09 × 10^9^/L), C + DS (11.58 + 2.08 × 10^9^/L), and C + DW (10.82 + 3.01 × 10^9^/L). The average for neutrophil cells is as shown in Figure 3. The highest neutrophil count was recorded by the C + LA500 group with an average of + S.E.M, (7.25 +1.87 × 10^9^/L), followed by the C + DS group, 4.68 + 1.30 × 10^9^/L, and the C + A group, 4.51 + 1.83 × 10^9^/L. The lowest volume was recorded by the C + LA250 group with an average + S.E.M, 4.30 + 0.59 × 10^9^/L. Lymphocyte count: The C + LA250 group recorded the highest lymphocyte count presented as average + S.E.M (8.95 + 1.391 × 10^9^/L), followed by the C + LA500 group (6.87 + 0.65 × 10^9^/L) and the C + DS group (5.86 + 0.75 × 10^9^/L). The lowest value was recorded by the C + A group, which was 5.36 + 1.06 × 10^9^/L.

### 2.5. Histological Results Showing LA Protected Inflammation from the Dermis of the Paw in Carrageenan-Induced Paw Edema in Rats

Figure 4 demonstrates the photomicrograph of the longitudinal section of the skin tissue structure of the rat’s hind paw for the model group (C + DW), positive control (C + DS), treatment group (C + LA250), and (C + LA500). The right photomicrographs represent the images captured with ×10 magnification. In these images, there is an abundance of infiltrating inflammatory cells residing throughout the tissue in the C + DW group. In the other groups (C + DS, C + LA250, and C + LA500), the number of inflammatory cells is less than the C + DW group. It was also found that there was a moderately clear separation of collagen tissue cells in the dermis layer of the rats’ hind paws in the C + LA250 group and mild separation of collagen tissue cells was seen in the C + DW group. The separation of collagen tissue indicates the development of edematous in the dermal layer tissue owing to inflammation.

The left photomicrographs show the images captured with ×40 magnification. It was found that the C + DW group shows heavy infiltration of inflammatory cells (polymorphonuclear leucocytes), which is indicated by the close distribution of cells despite sustaining tissue edema. The distribution of inflammatory cells is less congested in the C + DS and C + LA250 groups along with edema formation. The C + LA500 group, on the other hand, showed an accumulation of inflammatory cells almost the same as the C + DW group despite sustaining moderate edema formation.

### 2.6. LA Leaf Extract Potentially Reverses Pro-Inflammatory Cytokine Concentrations in Carrageenan-Induced Paw Edema in Rats

To evaluate the anti-inflammatory abilities of LA leaf extract in carrageenan-induced paw edema, the concentration of IL-1β, TNF-α, and IL-6 in the rats’ homogenate paws were quantified. The IL-1β and TNF-α levels from the model rat groups displayed increased concentrations compared to the positive control (C + DS) and treatment groups (C + LA250 and C + LA500) (Figure 5). One-way ANOVA showed statistically substantial variations in inflammatory cytokine concentrations in the brain of the various groups of rats.

The results showed that there was no significant difference in the IL-1β levels for the model C + DW group compared to the C + DS group (*p* = 0.488) and between the C + DW group compared to the C + LA500 group (*p* = 0.328.). However, IL-1β was significantly lower (*p* < 0.05) for the C + DW group compared to the C + LA250 group (*p* = 0.021). There was no significant difference in IL-1β between the C + DS group compared to the C + LA250 group (*p* = 0. 316) and the C + LA500 group (*p* = 0.990).

TNF-α of the hind foot in the C + DW, C + DS, C + LA250, and C + LA500 groups had an average SEM ± of 334.58 + 27.18 pg/mL, 251.87 + 32.30 pg/mL, 202.03 + 28.12 pg/mL, and 238.08 + 26.74 pg/mL, respectively. There was no significant difference in the TNF-α level in the model group (C + DW) compared with the C + DS and C + LA500 groups, (*p* = 0.207) and (*p* = 0.114), respectively. However, the TNF-α level had significantly lower results (*p* < 0.05) in C + LA250 compared to the model group (C + DW) with a value (*p* = 0.018). There was no significant difference in the TNF-α levels for the C + DS group compared to the C + LA250 group (*p* = 0.616) and the C + LA500 group (*p* = 0.986).

The mean levels of IL-6 pro-inflammatory cytokines in the plasma tissue of the hind paws of rats by group are as shown in Figure 5. The results of the study determined no significant changes between the four groups.

### 2.7. LA Leaf Extracts Potentially Reduce MDA Level and Increase TSOD Concentrations in Carrageenan-Induced Paw Edema in Rats

Figure 6 shows the ELISA results for MDA, GPX, and TSOD concentrations in the hind paws of the various rat groups. MDA levels of the hind foot: There was a decrease in MDA levels in the C + DS, C + LA250, and C + LA500 groups compared to the model group (C + DW) but it did not reach significant values. GPX enzyme activity: There was no significant difference observed between GPX1 activity among the groups. TSOD enzyme activity of the hind foot: All the positive control (C + DS) treatment groups (C + LA250 and C + LA500) showed significantly higher SOD activity compared to the C + DW group (*p* < 0.05). Both C + LA250 and C + LA500 also showed significantly higher SOD activities compared to the C + DS model group (*p* < 0.05). Statistically substantial variations in TSOD levels were detected among the various rat groups, as shown by one-way ANOVA. There were sharp variations in the increase in the TSOD level in the model group.

## 3. Discussion

Both endogenous and external substances can trigger inflammatory reactions, altering the vascular system and the cells of the affected tissues or organs. According to Patil et al. [34], carrageenan is a physiological agent that is not antigenic and does not exhibit notable systemic effects. The edema model, which involves inducing carrageenan on the hind legs of rats, was used for this study because of its great repeatability and extensive use in research involving inflammation.

All rat groups experienced an increase in edema as a result of the carrageenan induction that caused a phase of vascular changes, including transient vasoconstriction and vasodilation. Acute edema caused by the injection peaked after three hours. Previous studies have shown similar results, with the highest effect of carrageenan being well observed three hours after carrageenan injection and considered as the best time to study the anti-inflammatory activity of the tested products [35]. The 250 mg/kg and 500 mg/kg doses were obtained from the pilot study.

Diclofenac sodium as a NSAID drug was picked as the positive control group since it is effective in the treatment of inflammatory diseases like gout, rheumatoid arthritis, and osteoarthritis. It has the effect of inhibiting the action of COX enzyme and this leads to the reduction in production of prostaglandins. Almost all NSAID is absorbable orally but has a first-pass metabolism so that it enters systemic circulation in a proportion of 50 percent. The oral half-life is 1.12 h, and the mean residence time is 2.7 h [36,37,38]. Meanwhile L alata, a plant known for its medicinal properties, modulates the expression of inflammatory cytokines and mediators such as TNF-α, IL-1β, and NF-κB, thereby exerting anti-inflammatory and pain-relieving effects. The modulation of these cytokines is crucial in controlling inflammation and pain, as they are key players in the inflammatory response. The mechanisms by which *L. alata* achieves these effects can be understood by examining similar pathways and compounds in other studies [39,40].

The results of this study showed that the positive control group (C + DS) demonstrated the highest inhibition rate between the second and fourth hours, with values of 50.58% and 49.01%, respectively which correspond to the previous studies that showed the diclofenac sodium (10 mg/kg) had the greatest anti-inflammatory effects compared to the natural products being studied during the first four hours [41,42]. Beginning on the fourth, sixth, eighth, and twenty-four hours, the average plantar edema of the treatment groups (C + LA250 and C + LA500) was significantly lower than that of the model group (C + A). In comparison to the positive control group (C + DS), the treatment group (C + LA250) recorded the highest percentage of inhibition at the 24 h mark (63.25%), surpassing C + KA500 with a significant difference (*p* < 0.05). This suggests that the administration of *L. alata* aqueous extract was effective in reducing the swelling of paw edema caused by carrageenan, with 250 mg showing better effects compared to 500 mg.

The aqueous extract of *L. alata*, 250 mg/kg, may have a longer-lasting but slower-acting anti-inflammatory effect compared to diclofenac sodium. By inhibiting cyclooxygenase, the aqueous extract of *L. alata* most likely prevents prostaglandin synthesis during the second phase of inflammation. In a study of formalin-induced rat paw edema, Sprague Dawley rats were given the compound 2-himachalen-7-ol (7-HC) extracted from the bark of the Cedrus libani tree intraperitoneally. The results showed that a moderate dose of 50 mg/kg recorded a higher percentage of inhibition of rat paw edema than a dose of 100 mg/kg [43].

In comparison to the model group (C + DW), the treatment group’s infiltration score (C + LA250) was significantly lower. Histopathological examination revealed significant variations in the rats’ paw tissue between the C + DW and treatment groups (C + LA250 and C + LA500). The separation of collagen tissue cells in the subepidermal layer of the plantar tissue C + DW indicated widespread edema development. Additionally, there was a noticeable infiltration of leukocyte cells, particularly in the layer of dermis. The C + LA250 and C + LA500 treatments show a decrease in edema and inflammatory cell infiltration, particularly in the tissues of the dermis layer. In a study of carrageenan-induced rat foot inflammation, similar histological examination results were seen. At a dose of 50 mg/kg, the alkaloid extract from E. Cuneatum leaves demonstrated less infiltration and edema formation than the model group, which was administered with 2% Tween 20 [44] (Li et al., 2020). In comparison to the model group, the white blood cell, neutrophil, and lymphocyte counts of the rats receiving 250 mg/kg and 500 mg/kg *L. alata* extract were higher. This may be due to the peak time of inflammation occurring on the paws of carrageenan-induced rats at the fourth hour. However, the number of these three blood cells were not very prominent.

Cytokines are small proteins secreted by various cells, primarily immune cells, that play a crucial role in cell signalling. They are essential for modulating immune responses, inflammation, and hematopoiesis (formation of blood cells). Cytokines act as messengers between cells, coordinating the body’s response to infection, injury, and disease. Therefore, the anti-inflammatory activity of LA in this study was monitored by assessing the levels of pro-inflammatory cytokines such as IL-1β, IL-6, and TNF-α, which would increase in parallel with the inflammatory response. Based on the results of the IL-1β and TNF-α study of rat foot tissue, both treatment groups (C + LA250 and C + LA500) recorded lower value compared to the model group (C + DW) and the positive control group (C + DS). However, the C + LA250 was the only group showed a significant reduction in both IL-1β and TNF-α. IL-1β has important homeostatic functions, such as in the regulation of nutrition, sleep regulation, and human body temperature. In addition, IL-1β and TNF-α are involved in neutrophil migration in carrageenan induction inflammatory models. In this study, the treatment with LA did not show any significant effects to IL-6. However, the groups that received 250 mg recorded the lowest IL-6 values amongst all.

From the results of these inflammatory cytokine studies, it can be suggested that the administration of LA extract at both doses of 250 mg/kg and 500 mg/kg could inhibit inflammatory reactions. It is possible that nuclear factor kappa-light-chain-enhancer of activated B (NF-κB), a transcription factor involved in the acute inflammatory mechanism, was inhibited by LA. NF-κB acts by regulating the expression of cytokines such as TNF-α, IL-6, and IL-1, which are pro-inflammatory mediators.

The Rat Grimace Scale score is a non-invasive method for detecting the pain level of rats. This score has been validated as a method of assessing the level of pain, by assessing facial expressions related to the nature of pain. The higher the RGS score value, the higher the intensity of pain. In a laboratory environment, identifying, minimizing, and avoiding pain to animals is an ethical behaviour to improve animal welfare. Carrageenan induction inflammation caused a significant increase in the RGS score at the sixth to ninth hour after the injection [45]. From the results of the behavioural study conducted on the carrageenan induction model group of rat paws, there was an increasing trend of RGS scores that reached the highest score at the fourth to eighth hour. Supposedly, the results of this RGS score are in line with the results of the increase in the average edema of the rats’ feet measured physically. Nevertheless, at the 24th hour, the treatment group that received 250 mg/kg and 500 mg/kg of LA extract recorded higher RGS scores compared to the model group. The results of this score are not in line with the results of the study of the average edema of the rats’ paws measured physically at 24 h. Although the use of RGS scores has been confirmed to be able to assess pain in laboratory rats, these scores have some limitations. There are environmental factors that can affect the RGS score such as the way laboratory rats were handled throughout the study and the presence of the researcher during the behavioural study. The results of the other study on the sick behaviour of rats showed that there was an increase in the Mouse Grimace Scale score in rats that were controlled by lifting the tail compared to being lifted by the cupping method [46].

Inflammation induced by carrageenan has resulted in negative (C + DW) amplification of the MDA rate of the plantar homogeneous tissue of the model group. Although the changes in the MDA levels were not significant, both treatment groups (C + LA250 and C + LA500) recorded lower MDA values compared to the (C + DW) group. A lower MDA value reflects the low production of free radicals through lipid peroxidation. SOD enzymes convert superoxide radicals into hydrogen peroxide and oxygen molecules, while catalase and peroxidase enzymes convert the resulting hydrogen peroxide to water. In this way, two toxic species, superoxide radicals and hydrogen peroxide, can be converted into harmless end products. Carrageenan injections have been demonstrated to cause oxidative stress in rat paw tissue and the administration of LA extract can reduce oxidative stress as shown by the significant increase in the SOD activities in both groups that received LA extract. The increase in SOD activity was also significantly higher compared to the rats that received diclofenac sodium (C + DS). The low value of GPX-1 activity in the treatment group (K + LA250) is likely due to more enzymes acting as free radical scavengers produced during the inflammatory process. These results showed that there were antioxidant activities involved in the mechanism of LA action, which is absent in diclofenac sodium.

From this, it can be finalized that the administration of *L. alata* aqueous extract at a dose of 250 mg/kg can reduce the swelling of the paws of carrageenan-induced rats in a non-dose-dependent manner [43].

## 4. Materials and Methods

This study was approved by the Medical Research Ethics Committee of Universiti Kebangsaan Malaysia Medical Centre (grant code: GGPM-2019-008) and was conducted in accordance with the guidelines set by the Animal Ethics Committee, Universiti Kebangsaan Malaysia (approval number: ANAT/FP/2020/MOHD AMIR/14-MAY/1089-MAY-2020-SEPT.-2021). A total of 24 healthy Wistar breed male rats, weighing around 200–300 g, were taken from the Animal Resources Unit of the laboratory, Universiti Kebangsaan Malaysia, Kuala Lumpur Campus. The animals were housed in the animal laboratory, Preclinical Building, Universiti Kebangsaan Malaysia. The animals were placed in cages individually, maintained in a standard setting, i.e., 22 ± 22 °C, in a day and night cycle for every 12 h, and given rat food pellets and drink tap water ad libitum. The rats were left to adapt for 7 days before the study began.

### 4.1. Lepisanthes alata Leaves Extract

The LA plant was taken at the Forest of Knowledge Park, University of Malaya. This species has been identified and has been reposed at the University of Malaya Herbarium with a KLU50197 voucher number. LA extraction was carried out in the extraction laboratory of the Department of Pharmacology of Universiti Kebangsaan Malaysia.

#### 4.1.1. DPPH Radical Scavenger Activity

To each tube, 1 mL of the aqueous extract of LA was diluted with 10 mL of distilled water. Thereafter, aliquots of 0.1 mL of the mixture were prepared by adding it to 3.9 mL of an 80% ethanol solution of 0.6 mM DPPH·; ethanol (80%) was used as a blank solution while 3.9 mL of DPPH· + 0.1 mL of 80% ethanol was used as a control solution. The tube was agitated at high speed for 15 s and then left sedimenting for 180 min. The absorbance of each mixture was then taken at 515 nm using the spectrophotometer pall. This test was performed three times. The radical scavenging activity of DPPH was expressed in percentage of inhibition using the formula: inhibition = (A control −A sample) × 100)/A control.

#### 4.1.2. Total Phenolic Content

The total phenolic sample of the LA leaf aqueduct extract is determined using the modified Folin-Ciocalteu method. A total of 125 μL aliquots from each sample with concentrations of 0.2, 0.1, 0.15, and 0.05 mg were mixed with 0.5 mL of Folin-Ciocalteu reagents. The mixture is then vortexed for 15 s and left for six minutes. A total of 1.25 mL of 7% sodium carbonate solution is added, and then the solution is diluted by adding distilled water until it becomes 3 mL. After 90 min, the color of the solution changed, and the absorbance for each mixture was then measured at a wavelength of 760 nm using a spectrophotometer. The obtained values were compared with the standard curve of the gallic acid solution and expressed as the equivalent value of gallic acid in mg.

#### 4.1.3. Extraction

The LA leaf extraction process was carried out according to the previous study method [47] with some modifications. A total of 2.5 kg (wet weight) of LA leaves were taken and dried in an oven at 40 °C for 48 h. The dried leaves are weighed once again before they are ground into powder. LA leaf powder was then dissolved in distilled water at 40 °C in a ratio of 1:3 and then soaked for 30 m. The same step was repeated for three consecutive days. Each change of distilled water will be followed by filtration using Whitman paper No. 1. The filtrate was then stored in a −80 °C refrigerator before the freeze-drying process was carried out. The freeze-drying process was carried out for 72 h. The extract powder produced after the freeze-drying was stored at 4 °C in an airtight container for further use. The LA was weighed according to the dosage and dissolved in 1 mL of distilled water. The solution was then briefly shaken using a vortex machine before being administered through an oral force needle to the rats. The flow chart of the leaf extraction process of LA is shown in Figure 7.

### 4.2. Animal Study

A total of 24 male Wistar rats weighing 200–300 g was randomly divided into four groups, with each consisting of six rats. These rats were housed in special acrylic cages. Group I (C + DW) received carrageenan injection and distilled water orally; Group II (C + DS) received carrageenan injection and oral 25 mg/kg diclofenac sodium; Group III (C + LA250) received carrageenan injection and 250 mg/kg LA leaf extract orally; and Group IV (C + LA500) received carrageenan injection and 500 mg/kg LA leaf extract orally. Oedema on the right hindfoot of the study rats resulting from carrageenan was measured using a digital Vernier calliper at 0, 2, 4-, 6-, 8-, and 24-h post-carrageenan injection. The anti-inflammatory effect was assessed using the formula (1 − (Ct − C0) treatment group/(Ct − C0) model group) × 100%, where C0 is the average edema of the right hindfoot at 0 h and Ct is the average edema of the right hindfoot measured at a given time. At the same time, video recordings were taken using a camera for 10 min before the carrageenan injection, 30 min after the carrageenan injection, 30 min after the administration of oral treatment, and 30 min before the 2nd, 4th, 6th, 8th, and 24th hours. After 24 h, the rats were sacrificed using Ketamine, Zoletil-50 (Tiletamine and zolazepam) and Xylazine (KTX) 0.1 mL/100 g intraperitoneally. The right sole of the hind leg was harvested and divided into two parts. The first part was kept in 10% formalin for histological analysis, and the second and third parts were stored at −80 °C before pursuing protein analysis.

#### 4.2.1. Preparation of 1% Carrageenan

A total of 100 mL of 0.9% normal saline together with 1 g (1%) of carrageenan powder (Lambda Type IV, Sigma, Kanagawa, Japan) were mixed, stirred and heated at 90 °C for 5–10 min until the solution was well formed. The carrageenan solution was then allowed to cool at room temperature and stored in sterile glass bottles. A new solution was prepared for each use. 

#### 4.2.2. Measurement of the Effect of Oedema on the Hind Foot of Rats

Measurement of the effect of edema on the right hind foot of the rats was carried out according to the time specified in the study. The sole of the right hind foot to be injected was wiped with a 70% alcohol cotton swab before 0.2 mL of carrageenan solution was slowly injected into the subcutaneous layer of the skin using a 27 G 1/2″ needle. Change the sole edema of the right hind leg of a rat was carefully measured using a Vernier caliper so that it is not too tight so that it can hurt the rat. Measurement of the edema of the right hind foot of the rat was carried out according to the predetermined time in the study after the video recording was taken. 

#### 4.2.3. Assessment of Analgesic Behaviours Using the Rat Grimace Scale

The rats were placed in a specially constructed acrylic box (14 cm wide × 26.5 cm long × 20.5 cm high) with one transparent wall, two opaque side walls, and a fourth wall that was left open. The box was placed 1 m from the floor with the opening facing the floor. This arrangement created a visual cliff, and that encouraged the rats to face the camera. A camera was placed 0.24 m from the exposed wall to capture the video footage. This arrangement also allowed the video recording of four rats to be done simultaneously using two cameras.

Prior to the study, the rats were placed in the acrylic box for a maximum of 10 min for three consecutive days to familiarize them with space [45,48]. During the study, each rat was placed in its own observation box, and video recording was done for 10 min at each specific time. During the recording, the assessor left the room until the recording was completed. The video recordings were reviewed using Windows Media Player. Three still frames are taken manually to obtain the image, where all facial features were visible and not obscured by motion artifacts were selected from each 10-min video. These images were edited so that only the facial parts that included the rat’s eyes, ears, nose, and whiskers were clearly visible. The images were assigned with particular codes and fed into PowerPoint with each image placed on a single slide at random [45].

Analgesic response was carried out with the Rat Grimace Scale. There were four ‘facial action’ orbital entrapments: nose and cheek alignment, ear position, and rat whisker position were considered in this scoring system. A score of 0–2 was given where the score = 0 (none), score = 1 (present but not clear), and score = 2 (present and very clearly visible) [49].

#### 4.2.4. Blood Sampling and Isolation of Rat Serum

After 24 h, the rats were anesthetized by injection of cocktail solution KTX at the dose of 0.1 mL/100 g rats body weight via intra-peritoneal injection. Blood was drawn from the retro-orbital vein at an angle of 45° through the medial cantus of the rats, using glass capillaries into K2EDTA tube and delivered on the same day to the laboratory for full blood count (FBC). The rats were then euthanized by decapitation using rat guillotines.

#### 4.2.5. Harvesting Rat Hind Foot Tissue

The right hind legs of the rat were harvested. The hind soles were divided into two parts: for histology and enzyme-linked immunosorbent (ELISA) tests. The tissues for the histology study were stored in a 10% formalin solution, while the ELISAs were rinsed with a cooled phosphate buffer solution (PBS) and stored in a −80 °C freezer until further analysis.

#### 4.2.6. Histological Sample Preparation

Following euthanization, the animal was positioned dorsally, and the hind paw was cleaned with 70% ethanol. The Institutional ethical guidelines allowed researchers to obtain rat hind paws from euthanized animals. The researchers cleaned the tissue while making cuts before performing fixation using 4% paraformaldehyde at 4 °C. The fixation process (also including decalcification when needed) was followed by tissue dehydration with graded ethanol then xylene clearing prior to final paraffin embedment. A microtome sliced thin sections of 5–10 µm that got placed on slides for their gradual drying process. The appropriate stain selection for this study was hematoxylin and eosin (H&E). The laboratory concluded its procedure by using a light microscope to evaluate stained sections followed by documentation for analysis purposes.

#### 4.2.7. Preparation of Homogenate

The tissues of the right hind foot of the rats were weighed, cut into small pieces, and washed with PBS (0.01 M, pH 7.4). The tissue was dissolved in the RIPA buffer at the ratio of weight (g): volume of the RIPA buffer solution (ml) = 3:10. The tissue mixture and buffer solution were then at 5000 rpm for three cycles of 30 s each. The sample was left in ice for 30 min and centrifuged again at 12,000 rpm for 10 min at 4 °C, then the supernatant was taken and stored for the ELISA.

#### 4.2.8. Inflammatory Cells in Rat Paw Tissue

The histological sections of the rat paw were placed on a light microscopic platform. The examination was performed under 400× magnification for the optimum view. Photomicrographic images of the epidermis and dermis of rat paw tissue were taken randomly from 5 areas. The images were evaluated by two evaluators in a double-blind technique. The scoring was based as follows: 0 (no inflammation), 1 (mild inflammatory change), 2 (mild to moderate inflammatory change), 3 (moderate inflammatory change), 4 (moderate to severe inflammatory change) and 5 (severe inflammatory change) [50,51].

#### 4.2.9. Pro-Inflammatory Cytokines and Oxidative Stress Enzymes Studies

Inflammatory markers were analyzed by IL-1b, IL-6, and TNF-α ELISA kits (Elabscience, Houston, TX, USA) as directed by the manufacturer’s instructions. The absorbance was measured at 450 nm with a microplate reader (Thermofisher, Waltham, MA, USA). All samples were analyzed in triplicate.

Oxidative stress enzyme activities were analyzed using superoxide dismutase (T-SOD), glutathione peroxidase 1 (GPX1), and malondialdehyde (MDA) ELISA kits (Elabscience, Houston, TX, USA) as directed by the manufacturer’s instructions. The absorbance was measured at 450 nm with a microplate reader (Thermofisher, Waltham, MA, USA). All samples were analyzed in triplicate.

### 4.3. Statistical Analysis

The statistical analysis was conducted using version 26 of the IBM Statistical Package for the Social Sciences (SPSS). The mean + standard error of the mean (SEM) was the format used to display the gathered data. The parametric statistical test like one-way ANOVA and Tukey’s post hoc test were used for analysis of normally distributed data. In addition, for the repeated measurement, one-way repeated measure ANOVA was used. The non-parametric statistical test such as The Kruskal Wallis and Mann-Whitney post hoc tests were used for statistical analysis of non-normal data distributions and score data. A significant value was defined as a *p* value < 0.05.

## 5. Conclusions

From the results, it can be concluded that LA extract possesses anti-inflammatory and antioxidant properties that could reduce inflammation at the later stage of the inflammatory process. It has the potential to be used as an anti-inflammatory and analgesic agent, as well as in the prevention of acute and chronic inflammatory diseases.

## Figures and Tables

**Figure 1 pharmaceuticals-18-01142-f001:**
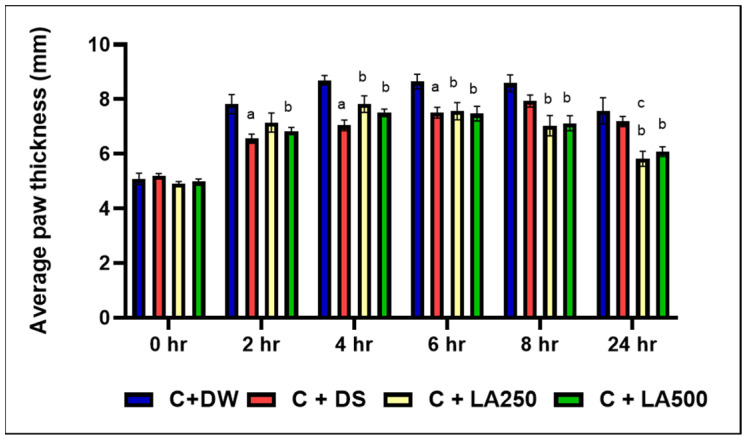
The average paw edema of the rats from the four groups was assessed at different time intervals during 0, 2, 4, 6, 8, and 24 h in the experiment. The model group in which the rats were treated with 1% carrageenan + distil water (C + DW), and a positive control group where rats in this group received 1% carrageenan + diclofenac sodium 25 mg/kg (C + DS). In addition, there were two different treatment groups, namely 1% carrageenan co-administered with LA 250 mg/kg (C + LA250) and LA 500 mg/kg (C + LA500), respectively. The values are shown as mean ± SEM; *n* = 6, (a. *p* < 0.05 positive control vs. model, b. *p* < 0.05 treatment vs. model, c. *p* < 0.05 treatment vs. positive control).

**Figure 2 pharmaceuticals-18-01142-f002:**
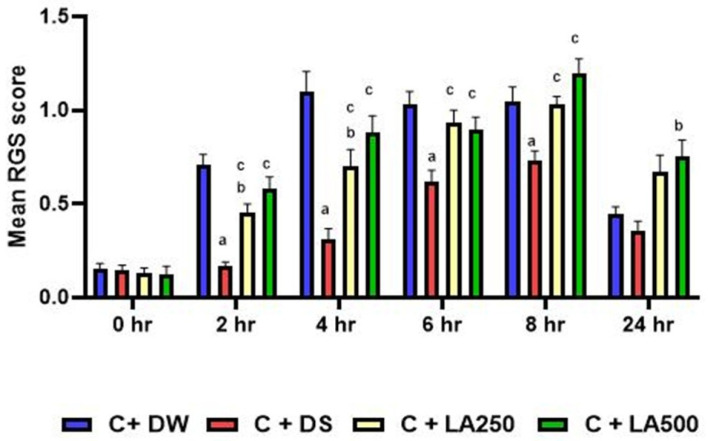
The average ‘Rats Grimace Scale’ (RGS) scores of the rats from the four groups were assessed at different time intervals during 0, 2, 4, 6, 8, and 24 h in the experiment. The model group had rats treated with 1% carrageenan + distil water (C + DW), and the rats in the positive control group received 1% carrageenan + diclofenac sodium 25 mg/kg (C + DS). In addition, there were two different treatment groups, namely 1% carrageenan co-administered with LA 250 mg/kg (C + LA250) and LA 500 mg/kg (C + LA500), respectively. The values are shown as mean ± SEM; *n* = 6, (a. *p* < 0.05 positive control vs. model, b. *p* < 0.05 treatment vs. model, c. *p* < 0.05 treatment vs. positive control).

**Figure 3 pharmaceuticals-18-01142-f003:**
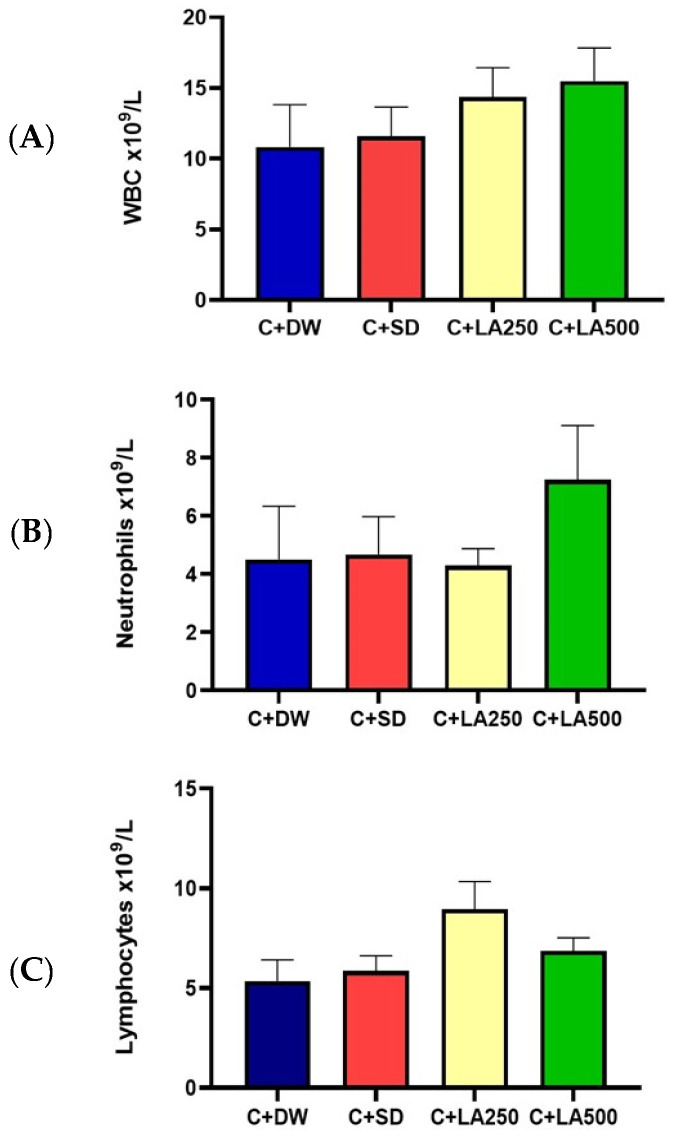
Bar charts to evaluate the white cells differential count. The experimental groups included the model group in which the rats were treated with 0.2 mls carrageenan injection 1% and distilled water orally (C + DW), and a positive control group where the rats received 0.2 mls carrageenan injection 1% and diclofenac sodium 25 mg/kg (C + DS). In addition, two different treatment groups, namely C + LA250 and C + LA500, received 0.2 mls carrageenan injection 1% + lepisathes alata leaf extraction 250 mg/kg and 500 mg/kg, respectively. (**A**) white cell count (WBC), (**B**) neutrophil count, and (**C**) lymphocyte count. The values are shown as mean ± SEM; *n* = 6.

**Figure 4 pharmaceuticals-18-01142-f004:**
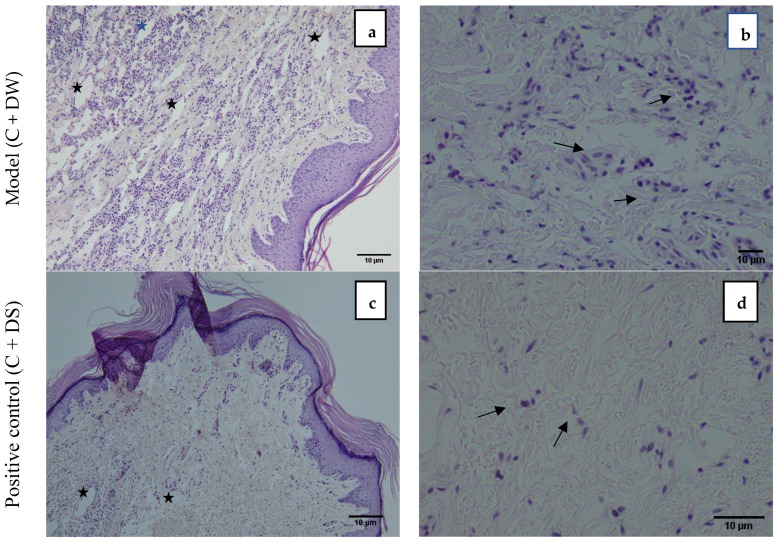
A group of photomicrographs from hematoxylin–eosin-stained paws illustrates how LA therapy affects the 1% carrageenan infiltration in rat compared to model rats (C + DW). The black arrows show the inflammatory cells. The asterisk symbols indicate edematous tissues. Right photomicrographs are ×10 while left photomicrographs are ×40 magnification, respectively. Scale bar: 10 µm. The experimental groups included the model group where the rats received carrageenan + distilled water (C + DW) and the positive control group in which the rats were treated with carrageenan + diclofenac sodium 25 mg/kg (C + DS). In addition, two different groups, C + LA250 and C + LA500, received carrageenan + LA extract leaf 250 mg/kg and carrageenan + LA extract leaf 500 mg/kg, respectively. (**a**,**b**) model C + DW, (**c**,**d**) positive control C + DS, (**e**,**f**) treatment C + LA250, (**g**,**h**) treatment C + LA500.

**Figure 5 pharmaceuticals-18-01142-f005:**
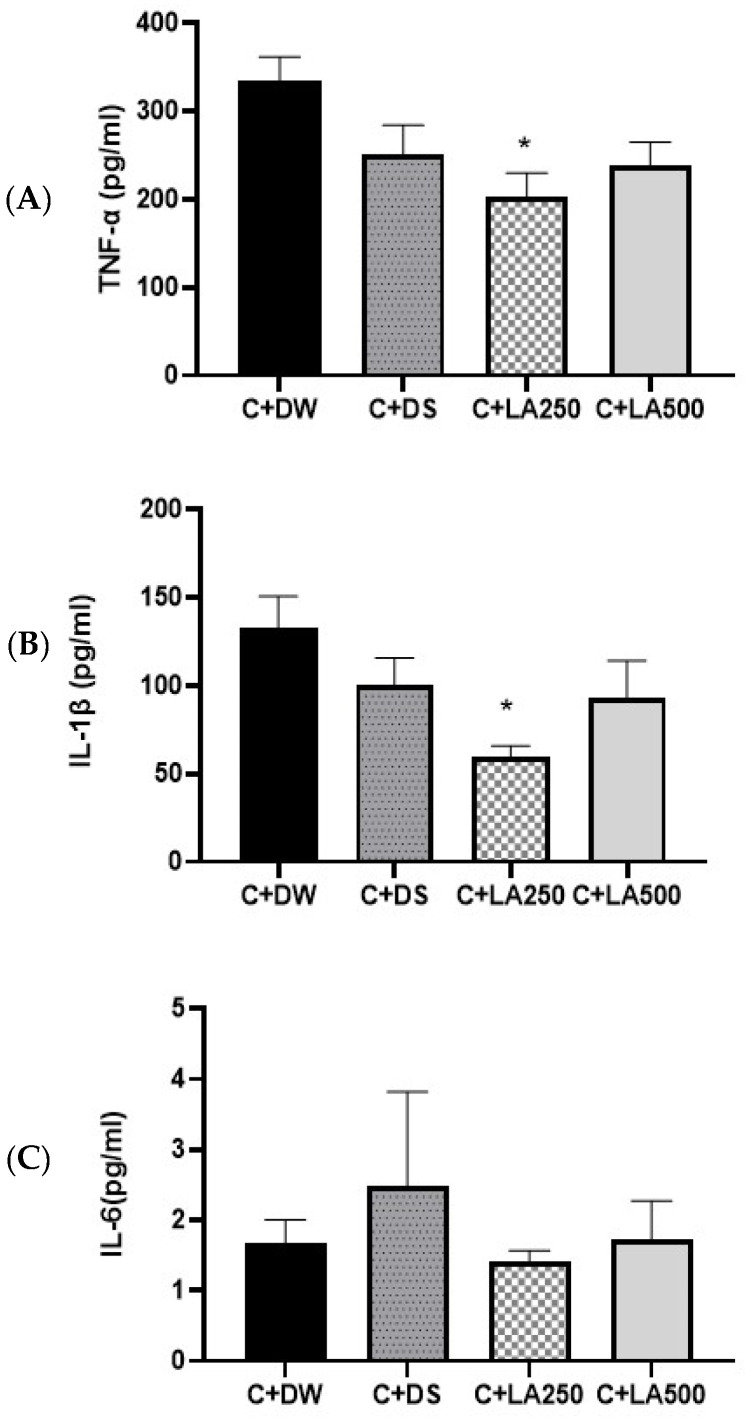
LA leaf extract lowers pro-inflammatory cytokine levels in carrageenan-induced paw edema in rats. A total of four groups were employed in the experiment. For the experimental model group, the rats were treated with carrageenan + distil water, and for the positive control group, the rats received carrageenan + diclofenac sodium 25 mg/kg. In addition, two different treatment groups, namely C + LA250 and C + LA500, received carrageenan + LA 250 mg/kg and carrageenan + LA 500 mg/kg, respectively. (**A**) TNF-α, (**B**) IL-1β, and (**C**) IL-6. The values are shown as mean ± SEM; *n* = 6, * *p* < 0.05 vs. model.

**Figure 6 pharmaceuticals-18-01142-f006:**
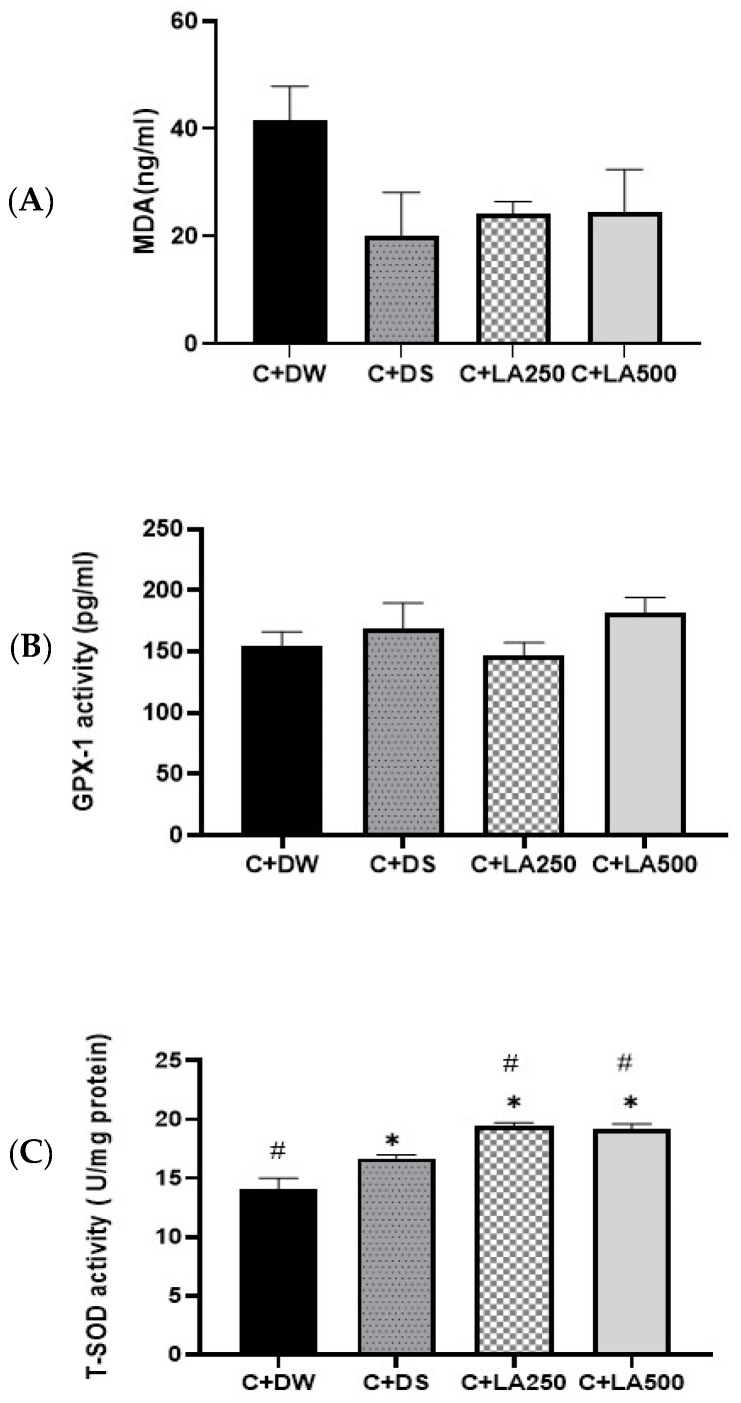
LA leaf extract lowers oxidative stress enzyme levels in carrageenan-induced paw edema in rats. A total of four groups were employed in the experiment. For the experimental model group, the rats were treated with carrageenan + distil water, and for the positive control group, the rats received carrageenan + diclofenac sodium 25 mg/kg. In addition, two different treatment groups, namely C + LA250 and C + LA500, received carrageenan + LA 250 mg/kg and carrageenan + LA 500 mg/kg, respectively. (**A**) MDA, (**B**) GPX1, and (**C**) TSOD. The values are shown as mean ± SEM; *n* = 6, * *p* < 0.05 treatments vs. positive control, # *p* < 0.05 treatments vs. model.

**Figure 7 pharmaceuticals-18-01142-f007:**
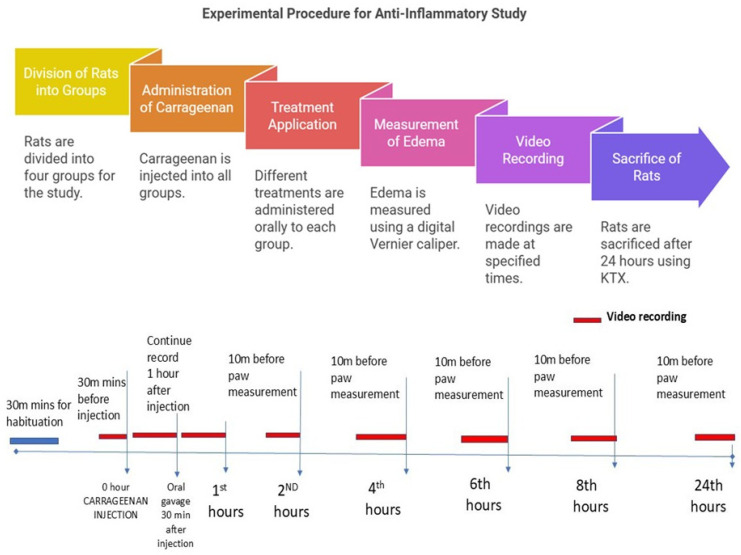
Experimental Procedure for Anti-Inflammatory Study with timeline.

## Data Availability

The original contributions presented in this study are included in the article. Further inquiries can be directed to the corresponding author.

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
