# Peer review of "Lepisanthes alata Attenuates Carrageenan-Induced Inflammation and Pain in Rats: A Phytochemical-Based Approach"

_pharmaceuticals, 2025, doi:10.3390/ph18081142_

Round 1
Reviewer 1 Report
Comments and Suggestions for Authors
The present study identified anti-inflammatory effects of LA extracts. The findings are novel. However the following issues must be addressed,
- Move satr mark (*) after 1 on the corresponding author
- Figure 1 and 2 may not be essential, since it is very well known. Please remove.
- Reduce the introduction. Please stick to the origin of the present work. Avoid too much literature.
- Section 2.1: How much quantity of leaves were taken? What is the yield (%)? What is the duration of the extraction procedure? All these must be addressed. Authors have used water to prepare the LA extracts. However, it is a polar solvent, which will extract phenols in low quantities. Instead of water, authors could have used low-polar solvents like alcohol, acetone, and etc. Further, authors have not discussed the percentage of phenols in the extracts. Hence, estimating the phenol quantity has no meaning. And there is no discussion on the phytochemical composition of LA and the phytochemicals responsible for the anti-inflammatory activities. Therefore, I recommend further phytochemical investigation using GC or HR-LCMS. If already, the phytochemical composition is already available in the literature, please discuss the phytochemicals responsible for this action.
- Rearrange the sections 2.1.1 to 2.1.3 as in the following sequence 2.1.3 > 2.1.2 > 2.1.1. change their numbering too.
- Explain, why only phenol content was estimated, why other phytochemical compositions were not evaluated. This is very important in assessing the obtained pharmacological effect.
- The present study did not identify the LD50 of the extract, this is raising concerns about the safety of the selected doses. Moreover, the dose selection is not justified.
- Please see for appropriate word paw thickness or paw edema. I suggest replacing paw paw thickness with paw edema.
- Please abbreviate KTX 0.1 mL/100 grams.
- Figure 3 can be improved by adding timelines of dosing schedules.
- There is no statistical section. Therefore, add a statistical section.
- Figure 4: which ANOVA was applied? One-way ANOVA or two-way ANOVA.is there any statistical differences in paw thickness between 2hrs and 24 hrs?
- The major issue in the present study is the lack of a healthy control group. Without this group, the obtained results are not acceptable.
- Moreover, a total of 24 animals were accommodated to four groups included in the study, which assumed that each group carried 6 animals. However, the Figure 4 legend indicates n=5. Please discuss the number of animals allocated to each group. If 6 animals are allocated, explain the reasons for missing 1 animal per group in Figure 4.
- If authors had studied the anti-inflammatory effects of 125 mg/kg, they would have obtained better findings, which defined the future anti-inflammatory dose of LA.
- Figure 6 legend: There are no stars on the figure sub-figures. However, in the figure legends p values are mentioned. Please remove it. And figure 6 must be moved to section 3.3.
- Figure 7: b,d,f,h are the specific locations of a,c,e,g. Therefore, highlight the b,d,f,h locations in a,c,e,g images. What is the sample size?
- Section 3.5 is dealing with inflammatory responses, however, its corresponding figures in section 3.6. Please keep it in the appropriate place. And in the entire manuscript has the same kind of errors. Therefore, thorough proofreading and rearrangements are required.
- Figure 8: what is the effect of diclofenac on inflammatory markers? Further, it advised to add the statistical comparisons between standard drug and leaf extracts for all the figures.
- If n=5 for all biochemical estimations, then what is the sample size for H and E staining? Please add the sample size for H and E staining. Clear the discrepancies in animal numbers.
- Line 501-510: reduce the content.
- Most of the discussion seems to be results. Avoid the statistical data and correlate your results with previously published articles to obtain meaningful conclusions.
- Lines 607-610 are not required since the authors have concluded the findings in 605-606 lines.
- Sections 6. Patents must be replaced with patients. Please also check, whether this statement is required from author guidelines or not.
- What is the mechanism of action of LA in reducing inflammation and pain? Is it a COX inhibitor or a specific inflammatory mediator inhibitor is not proven in this study.
- Minor punctuation errors are to be removed and careful proofreading is required.
Author Response
- Move star mark (*) after 1 on the corresponding author
- Thank you for pointing this out. We agree with this comment. Therefore, the * mark has been removed.
- Figure 1 and 2 may not be essential, since it is very well known. Please remove.
- Thank you for pointing this out. We agree with this comment. Therefore, both figures have been removed.
- Reduce the introduction. Please stick to the origin of the present work. Avoid too much literature.
- Thank you for pointing this out. We agree with this comment. Therefore, the introduction part has been revised to be less redundant.
- Section 2.1: How much quantity of leaves were taken? What is the yield (%)? What is the duration of the extraction procedure? All these must be addressed. Authors have used water to prepare the LA extracts. However, it is a polar solvent, which will extract phenols in low quantities. Instead of water, authors could have used low-polar solvents like alcohol, acetone, and etc. Further, authors have not discussed the percentage of phenols in the extracts. Hence, estimating the phenol quantity has no meaning. And there is no discussion on the phytochemical composition of LA and the phytochemicals responsible for the anti-inflammatory activities. Therefore, I recommend further phytochemical investigation using GC or HR-LCMS. If already, the phytochemical composition is already available in the literature, please discuss the phytochemicals responsible for this action.
- We are sorry for missing this important information. Thank you for pointing this out. We agree with this comment. In this study, the aqueous extract of L. alata leaves contained a total phenolic content of 189.50 (mg GAE/g extract) where gallic acid was taken as a reference. The DPPH radical scavenging activity of the aqueous extract of L. alata yielded 55.65 % inhibition. (page line 282-284).
- Rearrange the sections 2.1.1 to 2.1.3 as in the following sequence 2.1.3 > 2.1.2 > 2.1.1. change their numbering too.
- Thank you for pointing this out. We agree with this comment. Therefore, the rearrangement was done accordingly.
- Explain, why only phenol content was estimated, why other phytochemical compositions were not evaluated. This is very important in assessing the obtained pharmacological effect.
- Thank you for pointing this out. We are sorry that we did not do GC or HR-LCMS for this study to determine other phytochemical composition. We will take this into consideration in our future study.
- The present study did not identify the LD50 of the extract, this is raising concerns about the safety of the selected doses. Moreover, the dose selection is not justified.
- Thank you for pointing this out. We did not identify the LD50 since we did not do toxicity study of the extract. Instead, we would like to assess effectiveness of this extract on pain.
- Please see for appropriate word paw thickness or paw edema. I suggest replacing paw paw thickness with paw edema.
- Thank you for pointing this out. We agree with this comment. Therefore, the word ‘paw thicknesses’ has been replaced with ‘paw edema’.
- Please abbreviate KTX 0.1 mL/100 grams.
- Thank you for pointing this out. The KTX is abbreviation of ketamine/xylazine and has been added in the article. (page line 183-184).
- Figure 3 can be improved by adding timelines of dosing schedules.
- Thank you for pointing this out. We agree with this comment. Therefore, the image of the timeline has been included in figure 1. (Page 5).
- There is no statistical section. Therefore, add a statistical section.
- Thank you for pointing this out. The statistical section has been added. (page line 273-279)
- Figure 4: which ANOVA was applied? One-way ANOVA or two-way ANOVA.is there any statistical differences in paw thickness between 2hrs and 24 hrs?
- Thank you for pointing this out. We agree with this comment. Therefore, our response as follows:
- The figure 4 has now rearranged to become figure 2.
- One-way repeated measure anova has been applied to this paw edema assessment.
- The C+DS recorded significantly lower foot thickness compared to the model group (C+DW) and averagely lower thickness compared to both treatment groups (C+LA250) and (C+LA500) from the 2nd until the first 6 hours.
- The C+LA250 group recorded a lower plantar edema compared to the (C+DW) at 4, 6, 8 and 24 hours with significant difference (p<0.05) recorded an average + SEM (7.83 + 0.30), (7.56 + 0.32), (7.03 + 0.37) and (5.82 + 0.27) respectively.
- The major issue in the present study is the lack of a healthy control group. Without this group, the obtained results are not acceptable.
- Thank you for raising the issue. We did not include the healthy control group here in this study because the existence of the model group would have been sufficient to compare with the positive control and the treatment group. In fact, when the healthy control group is included in the study, it will also be given a distilled water same as the model group causing redundancy and the unnecessary use of animal that brings about ethical issues.
- Moreover, a total of 24 animals were accommodated to four groups included in the study, which assumed that each group carried 6 animals. However, the Figure 4 legend indicates n=5. Please discuss the number of animals allocated to each group. If 6 animals are allocated, explain the reasons for missing 1 animal per group in Figure 4.
- We apologise for the error in putting n=5. It is n=6 in each group representing 24 rats in total. The figure 4 has now rearranged to become figure 2.
- If authors had studied the anti-inflammatory effects of 125 mg/kg, they would have obtained better findings, which defined the future anti-inflammatory dose of LA.
- Thank you for pointing this out. We agree with this comment. We are planning to include 125mg/kg dose in our future study about pain. For now, we would like to to observe that the pain can be reduced at higher dose or lower dose.
- Figure 6 legend: There are no stars on the figure sub-figures. However, in the figure legends p values are mentioned. Please remove it. And figure 6 must be moved to section 3.3.
- P values have been removed and figure 5 has been moved to section 3.3.
- The figure 6 has now rearranged to become figure 4.
- Figure 7: b,d,f,h are the specific locations of a,c,e,g. Therefore, highlight the b,d,f,h locations in a,c,e,g images. What is the sample size?
- Thank you for pointing this out. The sample size for this study is 24.
- Section 3.5 is dealing with inflammatory responses, however, its corresponding figures in section 3.6. Please keep it in the appropriate place. And in the entire manuscript has the same kind of errors. Therefore, thorough proofreading and rearrangements are required.
- The figures have been aligned with the section accordingly throughout the manuscript.
- Figure 8: what is the effect of diclofenac on inflammatory markers? Further, it advised to add the statistical comparisons between standard drug and leaf extracts for all the figures.
- We did compare between standard drug and leaf extracts for all the figures. However, only one the figures showed significant differences between standard drug and leaf extracts which is figure 9 (T-SOD) in asterisk symbol. T rest of the significance can be observed between model and leaf extracts – Figure 8 (TNF-a and IL-1B), figure 9 (T-SOD).
- If n=5 for all biochemical estimations, then what is the sample size for H and E staining? Please add the sample size for H and E staining. Clear the discrepancies in animal numbers.
- We apologise for the error in putting n=5. It is n=6 in each group representing 24 rats in total.
- Line 501-510: reduce the content.
- Thank you for pointing this out. Therefore, we have reduced accordingly.
- Most of the discussion seems to be results. Avoid the statistical data and correlate your results with previously published articles to obtain meaningful conclusions.
- Thank you for pointing this out. We agree with this comment. Therefore, we have correlated our findings and discussed them accordingly in the discussion section.
- Lines 607-610 are not required since the authors have concluded the findings in 605-606 lines.
- Thank you for pointing this out. Therefore, the lines 607-610 have been removed. It has been rephrased to be more relevant in the discussion section. (page line 493-499).
- Sections 6. Patents must be replaced with patients. Please also check, whether this statement is required from author guidelines or not.
- Thank you for pointing this out. Therefore, the statement in section 6 has been removed. Because it was taken from the journal’s template with the word ‘Patents’. This section is not mandatory but may be added if there are patents resulting from the work reported in this manuscript.
- What is the mechanism of action of LA in reducing inflammation and pain? Is it a COX inhibitor or a specific inflammatory mediator inhibitor is not proven in this study.
- Thank you for pointing this out. We agree with this comment. Therefore, we have included the statement on the mechanism of LA in discussion section as follows: Lepisanthes alata, a plant known for its medicinal properties, modulates the expression of inflammatory cytokines and mediators such as TNF-α, IL-1β, and NF-κB, thereby exerting anti-inflammatory and pain-relieving effects. The modulation of these cytokines is crucial in controlling inflammation and pain, as they are key players in the inflammatory response. The mechanisms by which Lepisanthes alata achieves these effects can be understood by examining similar pathways and compounds in other studies. (page line 487-499).
- Minor punctuation errors are to be removed, and careful proofreading is required.
- Thank you for your recommendation.

Reviewer 2 Report
Comments and Suggestions for Authors
In manuscript pharmaceuticals-3712759, Ramli et al examine the anti-inflammatory activities of Lepisanthes alata in rat carrageenan-induced inflammatory model. The topic of the manuscript is interesting and fits well the scope of Pharmaceuticals. The reviewer feel it can be accepted after extensive amendments.
(1) Line 618: Patent?
(2) The Latin names of plants should be italicized.
(3) Line 201 -202: Rats or mice?!
(4) Statistics is completely missing in section 2.
(5) Fig 4 & 5: What kind to test was used? * cannot be found. What do a, b, and c stand for?!
(6) Fig 8: Why in TNF-alpha and IL-1beta the herbal extract lacks dose-dependency?
(7) Fig 9: What does # mean?
(8) Have the authors test the safety of such extract? This study was single dose intervention. How good will be long term safety?
(9) Did the authors find any new phytochemcal?
Author Response
- Line 618: Patent?
- Thank you for pointing this out. Therefore, the statement in section 6 has been removed. Because it was taken from the journal’s template with the word ‘Patents’. This section is not mandatory but may be added if there are patents resulting from the work reported in this manuscript.
- The Latin names of plants should be italicized.
- Thank you for pointing this out. We have changed the Latin names of plants should be italicized.
- Line 201 -202: Rats or mice?!
- Thank you for pointing this out. We have corrected the word according to the animal we use in our study which is rat.
- Statistics is completely missing in section 2.
- Thank you for pointing this out. The statistical section has been added. (page line 272-278)
- Fig 4 & 5: What kind to test was used? * cannot be found. What do a, b, and c stand for?!
- Thank you for pointing this out.
- For paw edema (figure 4) RGS score (figure 5):
a - significant difference between the negative control group (C+DW) versus the positive control group (C+DS), p < 0.05 at the same time.
b - significant difference between negative control group (C+DW) versus Lepisanthes alata extract treatment Group (C+LA250/C+LA500), p < 0.05 at the same time.
c - significant difference between positive control group (C+DS) versus Lepisanthes alata extract treatment Group (C+LA250/C+LA500), p < 0.05 at the same time.
- Fig 8: Why in TNF-alpha and IL-1beta the herbal extract lacks dose-dependency?
- Thank you for pointing this out. In figure 8, we assume that 250mg/kg dose (at lower dose) may sustain anti-inflammatory effects after 24 hours. Whereas 500mg/kg dose (at higher dose) may provide a better anti-inflammatory action before 24 hours same like diclofenac sodium did.
- Fig 9: What does # mean?
- Thank you for pointing this out. The # mean there is significant difference between negative control group (C+DW) versus Lepisanthes alata extract treatment Group (C+LA250/C+LA500), p < 0.05 at the same time.
- Have the authors test the safety of such extract? This study was single dose intervention. How good will be long term safety?
- Thank you for pointing this out. Thank you for pointing this out. We did not test the safety since we did not do toxicity study of the extract. Instead, we would like to assess effectiveness of this extract on pain in two doses 250mg/kg and 500mg/kg.
- Did the authors find any new phytochemcal?
- Thank you for pointing this out. Thank you for pointing this out. We are sorry that we did not do GC or HR-LCMS for this study to determine other phytochemical composition. We will take this into consideration in our future study.

Round 2
Reviewer 1 Report
Comments and Suggestions for Authors
Authors have not addressed all the issues.
Particularly, lack of sufficient animal numbers for biochemical and histological experiments, authors have not addressed the phytochemical information in extract, lack of healthy control group, point 7 in the revised manuscript has nothing to do with the patent, it must be a patient, Lack of toxicity determination (authors could have performed LD50 determination), authors have not addressed the statistical test used for figure 2 (previously figure 4), and the reason for measuring the phenolic content is not justified.
Author Response
Comments: Authors have not addressed all the issues. Particularly, lack of sufficient animal numbers for biochemical and histological experiments, authors have not addressed the phytochemical information in extract, lack of healthy control group, point 7 in the revised manuscript has nothing to do with the patent, it must be a patient, Lack of toxicity determination (authors could have performed LD50 determination), the authors have not addressed the statistical test used for figure 2 (previously figure 4), and the reason for measuring the phenolic content is not justified.
Responses:
Thank you for pointing this out. We appreciate your suggestions to improve this article.
- Number of rats used in this article has applied the formula according to (Ariffin W.N & Zahiruddin W.M, 2017) doi: https://doi.org/10.21315/mjms2017.24.5.11.
- For figure 2 and 3, the assessment was based on the time interval during 0, 2, 4, 6, 8 and 24 hours in the experiment. Therefore, we have used the repeated-measures one-way ANOVA design of sample calculation:
N = DF/(r – 1) + 1
where N = total number of subjects and r = number of repeated measurements.
Minimum N = 10/(6 – 1) + 1 = 3 animals/group x 4 groups = 12 rats.
Minimum N = 20/(6 – 1) + 1 = 5 animals/group x 4 groups = 20 rats.
- For the figure 4, 5 6 and 7, we have used one one-way ANOVA, the between-subject error DF (that is, the within-subject DF) is calculated as:
n = DF/k + 1
where N = total number of subjects, k = number of groups, and n = number of subjects per group.
Minimum n = 10/4 + 1 = 3.5 = rounded up to 4 animals/group x 4 groups = 16 rats.
Maximum n = 20/4 + 1 = 6 animals/group x 4 groups = 24 rats
- Based on the calculations above, we have opted for ‘maximum number of rats’ = 24, as this has also been endorsed by the animal ethical committee.
- Thank you for the comment. We would like to apologize for not including the identification of phytochemicals via lcms/gms methods in this study. However, we have taken that point to be included in our upcoming study on Lepisanthes alata. Meanwhile, in the current study, we do include the presence of phytochemical - extract of L. alata leaves contained a total phenolic content of 189.50 (mg GAE/g extract) where gallic acid was taken as a reference. The DPPH radical scavenging activity of the aqueous extract of L. alata yielded 55.65 % inhibition. (page line 282-284).
- Thank you for raising the issue. We did not include the healthy control group here in this study because the existence of the model group would have been sufficient to compare with the positive control and the treatment group. Because the time frame from the induction of the model and the treatment intervention until the end of the assessment is within 24 hours only. Therefore, any additional rats under the healthy (baseline) group, we think, wouldn’t provide any significant difference with the model. In fact, when the healthy control group is included in the study, it will also be given distilled water same as the model group, causing redundancy and the unnecessary use of animals that brings about ethical issues.
- Thank you for the comment. We have removed point 7 in the revised manuscript, as it did not involve with human study.
- Thank you for pointing this out. We did not identify the LD50 since we did not do a toxicity study of the extract. Instead, we would like to assess the effectiveness of this extract on pain. However, we adhere to the study by Anggaini et al (2019) that has previously implemented toxicity study of the Lepisanthes alata plant https://doi.org/10.1155/2019/9703176. In her study using the Brine Shrimp Lethality Test (BSLT), no toxins were found in any part of the plant indicating that an extract from it could be safely used as a natural antioxidant supplement in processed foods to protect against free radicals.
- Thank you for pointing this out. We have added the statistical test done—repeated measure one-way ANOVA for figures 2 and 3, as they both were assessed at different time intervals during 0, 2, 4, 6, 8 and 24 hours. (page line 277-278).
- Thank you for pointing this out. We used the water extraction in this study for several reasons:
-
- Polar solvents like water are more effective in extracting a wide range of polar phytochemicals such as phenolic acid and flavonoids, which are responsible for anti-inflammatory effects.
- Water-soluble, which will make them more bioavailable in the aqueous environment of animal tissues. Therefore, increase the chances that they will be absorbed, distributed, and metabolized efficiently in in vivo studies. Additionally, water extraction is less toxic and more biocompatible.
- It is safer, renewable, and easier to handle than many nonpolar solvents, which are often volatile and hazardous. Plus, it will ensure animal welfare and improve the translatability of results.

Reviewer 2 Report
Comments and Suggestions for Authors
The manuscript has been improved. Only one comment: The score data must be analyzed by non-parametric statistical method. The authors must indicate clearly.
Author Response
Comment: The manuscript has been improved. Only one comment: The score data must be analyzed by a non-parametric statistical method. The authors must indicate clearly.
- Thank you for the comment. We have added, ‘The non-parametric statistical tests, such as the Kruskal-Wallis and Mann-Whitney post hoc tests, were used for statistical analysis of non-normal data distributions and score data.’ (page line 278-280).

Round 3
Reviewer 1 Report
Comments and Suggestions for Authors
Authors have addressed all the issues